# Multifaceted Role of Apolipoprotein C3 in Cardiovascular Disease Risk and Metabolic Disorder in Diabetes

**DOI:** 10.3390/ijms252312759

**Published:** 2024-11-27

**Authors:** Bo-Yi Pan, Chen-Sheng Chen, Fang-Yu Chen, Ming-Yi Shen

**Affiliations:** 1Graduate Institute of Biomedical Sciences, China Medical University, Taichung 40402, Taiwan; u106010831@cmu.edu.tw (B.-Y.P.); fyc0321@gmail.com (F.-Y.C.); 2The Ph.D. Program for Cancer Biology and Drug Discovery, China Medical University and Academia Sinica, Taichung 40402, Taiwan; u105071204@cmu.edu.tw; 3Department of Medical Research, China Medical University Hospital, Taichung 40402, Taiwan; 4Department of Nursing, Asia University, Taichung 413305, Taiwan

**Keywords:** apolipoprotein C3, lipoprotein, atherosclerosis, cardiovascular diseases, diabetes mellitus, biomarker

## Abstract

Apolipoprotein C3 (APOC3) plays a critical role in regulating triglyceride levels and serves as a key predictor of cardiovascular disease (CVD) risk, particularly in patients with diabetes. While APOC3 is known to inhibit lipoprotein lipase, recent findings reveal its broader influence across lipoprotein metabolism, where it modulates the structure and function of various lipoproteins. Therefore, this review examines the complex metabolic cycle of APOC3, emphasizing the impact of APOC3-containing lipoproteins on human metabolism, particularly in patients with diabetes. Notably, APOC3 affects triglyceride-rich lipoproteins and causes structural changes in high-, very low-, intermediate-, and low-density lipoproteins, thereby increasing CVD risk. Evidence suggests that elevated APOC3 levels—above the proposed safe range of 10–15 mg/dL—correlate with clinically significant CVD outcomes. Recognizing APOC3 as a promising biomarker for CVD, this review underscores the urgent need for high-throughput, clinically feasible methods to further investigate its role in lipoprotein physiology in both animal models and human studies. Additionally, we analyze the relationship between APOC3-related genes and lipoproteins, reinforcing the value of large-population studies to understand the impact of APOC3 on metabolic diseases. Ultimately, this review supports the development of therapeutic strategies targeting APOC3 reduction as a preventive approach for diabetes-related CVD.

## 1. Introduction

In the course of diabetes, the risk of cardiovascular diseases (CVDs) increases by 1.38 times, and mortality rates rise by 1.86 times [1]. However, many studies show that merely controlling blood glucose levels in patients with diabetes is insufficient in effectively improving CVD risk [2,3]. Patients with diabetes are more susceptible to the development of CVD owing to a number of factors besides glucose, including smoking, high blood pressure, dyslipidemia, obesity, abdominal fat, chronic renal disease, and a family history of heart disease [4]. Several studies have suggested that apolipoprotein C3 (APOC3, Figure 1) plays a significant role in the development of CVD in patients with diabetes [5,6].

Cardiometabolic disease refers to a shared pathophysiological process that contributes to both metabolic diseases and CVDs [7]. Packard et al. have highlighted the pivotal role of APOC3 in cardiometabolic health [8]. Among the various contributors to CVDs, atherosclerosis—primarily driven by dyslipidemia—stands out as a critical factor [9,10]. Dyslipidemia has been firmly established as a key cause of atherosclerosis and subsequent CVDs [11]. Epidemiological and Mendelian randomization studies have demonstrated that reducing low-density lipoprotein cholesterol (LDL-C) levels significantly lowers the incidence of CVDs [12]. However, despite effective LDL-C control, adverse cardiovascular events remain a major clinical challenge [13]. While LDL-C is a well-recognized driver of atherosclerosis and CVD, it does not fully explain cardiovascular risk [14]. Increasing evidence underscores the significant role of triglycerides (TGs) and TG-rich lipoproteins (TRLs) towards cardiovascular risk [15]. ApoC3, a critical regulator of TG metabolism, has been identified as a key player in this process. Mendelian randomization studies have revealed that loss-of-function mutations in the *APOC3* gene are associated with reduced plasma TG levels and a lower risk of coronary heart disease [14]. Moreover, emerging evidence suggests that APOC3 serves as a critical risk factor for increased incidence of CVDs, with TRLs contributing to the progression of atherosclerotic plaque more directly than TGs themselves [16]. Although high-density lipoprotein (HDL) and LDL are not classified as TRLs, numerous studies have demonstrated that the presence of APOC3 in HDL and LDL is significantly associated with CVDs. This finding highlights the multifaceted roles of APOC3 and its implications for cardiovascular health, making this phenomenon especially noteworthy.

In addition, we have specifically organized the role of APOC3 in very low-density lipoprotein (VLDL), LDL, and HDL. APOC3 not only plays a role in TRLs but also plays an important role in these different types of lipoproteins. Beyond its involvement in lipid metabolism, APOC3 is crucial for regulating TG levels and modulating lipoprotein lipase (LPL) activity, both of which are key components in the pathogenesis of diabetes-associated CVDs [17]. Therefore, understanding how APOC3 affects the metabolism of TRLs and contributes to diabetes-related CVD is essential for developing potential therapeutic strategies, including monoclonal antibodies and small interfering RNA (siRNA) therapies aimed at reducing the APOC3 levels [18]. In addition to pharmacological interventions, lifestyle modifications, such as diet and exercise, have been shown to influence APOC3 expression and TRL levels, providing a comprehensive approach to managing cardiovascular risk in diabetes [19,20].

Therefore, this review aims to provide an overview of the role of APOC3 in regulating cardiometabolic risk, particularly its effects on TG metabolism, and narrow the focus to its specific involvement in diabetes-associated CVDs. The review will also highlight the latest research on APOC3 in diabetic populations, including longitudinal studies tracking its levels over time, in order to offer a more integrated understanding of its contribution to the cardiovascular risk in diabetes.

## 2. Diabetes and Cardiovascular Disease Connection: The Influence of APOC3

The etiologies of CVDs and types 1 and 2 diabetes mellitus (T1DM and T2DM, respectively) are closely associated with APOC3 [5,21,22]. APOC3, an 8.8 kDa protein composed of 79 amino acids, is glycosylated at threonine-74 [23] and encoded by a gene located on chromosome 11q23 [23] (Figure 1). The protein is predominantly expressed in the liver and, to a lesser extent, in the small intestine [24]. APOC3 can exist in three isoforms—C-III0, C-III1, and C-III2—based on sialic acid residues [25]. These isoforms vary in mass and pH [26], and their relative plasma levels impact metabolic effects: apoC-III1 and apoC-III2 account for 55% and 40%, respectively, while apoC-III0 is present only in trace amounts [27]. Functionally, APOC3 opposes APOC2 and inhibits LPL transferred to HDL during TRL lipolysis [17]. The primary action of LPL is the gradual hydrolysis of TRL in chylomicrons and very LDLs (VLDLs) into LDLs [28] (Figure 1). Lowering APOC3 levels has been shown to reduce TG levels in humans and animal models [29], and elevated APOC3 levels are strongly correlated with hypertriglyceridemia [29,30] and an increased CVD risk.

APOC3 expression is modulated by multiple regulatory factors (Figure 2). In response to insulin, hepatic FOXO1 production decreases, resulting in higher APOC3 expression, which elevates plasma TG levels and impairs lipid tolerance [31]. Glucose also induces APOC3 transcription in hepatocytes via carbohydrate response element-binding protein and hepatocyte nuclear factor 4⍺ pathways [32]. After meals, free fatty acids activate peroxisome proliferator-activated receptor (PPAR)-γ, while PPAR-γ coactivator 1β (PGC-1β) facilitates hepatic lipogenesis and lipoprotein secretion, leading to hyperlipidemia [33]. Therefore, PGC-1β serves as a critical regulator of APOC3 expression [34]. In diabetes, decreased insulin, elevated saturated fatty acids, and carbohydrate surges (e.g., glucose and fructose) collectively upregulate APOC3, increasing TG levels and TRLs accordingly. Diabetes is usually accompanied by dyslipidemia, a key contributor to CVD [35]. TGs are independent risk factors for both T1DM and T2DM [36,37], with APOC3 as a primary TG regulator [38]. Insulin and glucose directly influence APOC3, as variation in insulin-responsive elements in the APOC3 promoter affects its expression depending on insulin and glucose concentrations [39]. APOC3 messenger RNA levels have been shown to rise by 1.4- to 1.5-fold in APOC3-high, insulin-deficient mice, whereas insulin administration reduces hepatic APOC3 expression [40]. Additionally, APOC3 may participate in the inflammatory response in both T1DM and T2DM, positioning APOC3 as a potential target in diabetes management. Factors such as insulin resistance, dysregulated lipoprotein metabolism, and chronic inflammation elevate APOC3 levels in patients with diabetes. Insulin resistance, coupled with hyperglycemia and insufficient insulin, has been shown to raise APOC3 concentrations [5,41]. Lipoprotein dysregulation increases hepatic VLDL and serum TGs [42], while chronic inflammation and unregulated free fatty acid re-esterification further elevate TGs [43]. Together, these factors disrupt lipid metabolism and exacerbate CVD risk in diabetic populations, underscoring the critical role of APOC3 in managing diabetes-related CVDs.

### 2.1. Type 1 Diabetes Mellitus

The pathological cause of T1DM is attributed to the immunological damage to the islet cells of the pancreas, which impairs normal insulin secretion [45]. Hereditary and uncertain environmental factors also account for the pathogenesis [46]. Moreover, the later phases of T1DM may lead to the risk of CVD [47]. According to the International Diabetes Federation’s most recent estimate, 1.1 million children and teenagers aged <20 years have T1DM [48]. In a study, coronary artery calcification in type 1 diabetes, insulin deficiency was found to increase the level of APOC3, accelerate the formation of macrophage foam cells, and contribute to atherosclerosis; therefore, the expression of APOC3 can predict the risk of CVD in T1DM [5]. High levels of APOC3 are detected in patients with T1DM [49]. Mouse islet cells stimulated with APOC3 reportedly showed an increase in the concentration of cytoplasmic free intracellular Ca^2+^, resulting in islet β-cell death [49]. The Epidemiology of Diabetes Interventions and Complications study [50], which involved a cohort of patients with T1DM (*n* = 465), found that serum total APOC3 and APOC3-containing HDL were significantly associated with CVD in T1DM. In that study, an increase of 1 mg/dL in APOC3 is associated with a 10% higher incidence of CVD [50]. Higher APOC3 concentrations were found to accelerate the progression of the disease in a mice model with T1DM. Additionally, the overexpression of APOC3 triggered the activation of renal toll-like receptor 2 and nuclear factor κappa B (NF-κB) signaling pathways as well as elevated renal gene expression levels of downstream inflammatory factors such as tumor necrosis factor ⍺ (TNF-⍺), vascular cell adhesion molecule (VCAM)-1, and monocyte chemoattractant protein 1 [21]. In summary, although the etiology of T1DM is attributable to defective immune responses, T1DM-associated CVD is one of the key causes of death, and APOC3 can cause damage to pancreatic islet cells and kidneys [41].

### 2.2. Type 2 Diabetes Mellitus

Insulin resistance and β-cell dysfunction are the causes of T2DM [51]. A study of *ob*/*ob* mice showed that insulin resistance accelerated the secretion of APOC3, which consequently caused inflammation of the islet cells as well as increased mitochondrial metabolism, free cytoplasmic calcium ions, and apoptosis [51]. A meta-analysis evaluating the impact of body mass index (BMI) and APOC3 promoter variants (−482C > T and −455T > C) on the risk of T2DM (*n* = 7983) showed that the 482T allele was linked to a higher risk of T2DM prevalence and incidence. According to van Hoek et al., the odds ratios (ORs) for −482CT and −482TT were 1.47 (95% confidence interval [CI], 1.13–1.92) and 1.40 (95% CI, 0.83–2.35), respectively, with *p* = 0.009 for trend, and the hazard ratios were 1.35 (95% CI, 0.96–1.89) and 1.68 (95% CI, 0.91–3.1), respectively, with *p* = 0.03 for trend [52]. On the contrary, in these groups, the −455C allele seems to have a protective effect, reducing the risk of type 2 diabetes [52]. End-stage renal failure is a hallmark of end-stage T2DM [53]. Many studies have shown that kidney disease is related to the expression of APOC3 [54]. According to Hu et al., a significant (*p* < 0.01) increase was noted in APOC3 levels across the course of the disease in patients with T2DM and T2DM-associated nephropathy compared with normal individuals [55]. In a study, a set of 60 healthy participants (male, *n* = 30; female, *n* = 30) aged 47–65 years were selected as controls, whereas 120 patients with T2DM, aged 48–66 years, were recruited from Hebei General Hospital (Shijiazhuang, China) [55]. A strong correlation exists between circulating proprotein convertase subtilisin/kexin type 9 and inflammation markers PTX3 and APOC3, which may be the cause of atherosclerosis in T2DM [56]. Change in glucose level has been shown to be related to the secretion rate of APOC3; however, after liraglutide treatment, the concentration of APOC3 could be reduced by approximately 1.7 ± 0.5 mg/dL (*p* = 0.035) [57]. Volanesorsen is an antisense inhibitor of APOC3 and is believed to increase insulin sensitivity by 57%; furthermore, it reduces fructosamine, glycated hemoglobin, and glycated albumin levels and can be used to treat T2DM [58].

## 3. Mechanistic Roles of APOC3 in Disease Pathology

### 3.1. APOC3 Gene Variants and Regulation

APOC3 has multiple genetic variants, which may have significant implications for clinical treatment. The *APOC3* gene is located in a gene cluster with APOA1/APOA4/APOA5 linked to human chromosome 11, which increases the difficulty of identifying and characterizing APOC3 genetic variants [59]. An *APOC3* variant (rs138326449-A; minor allele frequency, 0.25% [UK]) has been reported to be strongly correlated with lower TG levels, with reductions ranging from 37% to 44% (*p* < 0.001); HDL levels; and reduced risk of ischemic CVDs [60,61]. Humans with the *APOC3* variant (rs138326449-A) show reductions in many metabolites involved in the triacylglyceride and acyl-acyl glycerophospholipids (both acyl-acyl and acyl-alkyl) synthesis pathways [62]. Conversely, another variant of *APOC3* (rs5128) was strongly associated with higher levels of APOC3, TG, total cholesterol (TC), and LDL-C [63]. *APOC3* (rs5128) was linked to higher BMI and an increased risk of dyslipidemia and obesity, with an OR of 4.022 (95% CI, 1.13–14.30) in a Kuwaiti Arab population study [64]. By promoting micro-RNA-4271 binding, the T allele of rs4225 inhibits APOC3 translation and may lower the risk of coronary heart disease (CHD) [65]. In summary, APOC3 is involved in the circulation of blood lipids, and *APOC3* gene loci mutations alter BMI. Moreover, an in-depth analysis of the standard position of the *APOC3* gene may address dyslipidemia and provide a cure for CVDs.

### 3.2. Functional Roles of APOC3 Across Lipoproteins

#### 3.2.1. High-Density Lipoprotein

Research into the physiological role of APOC3-containing HDL is still emerging. While APOC3 is recognized primarily for inhibiting plasma LPL and regulating plasma TG levels, it also plays a distinct role in the metabolism of TRLs within atherosclerotic vessels. This contrasts with the traditional view of HDL as protective against atherosclerosis, as APOC3-containing HDL has been associated with a negative impact on cardiovascular health, with studies indicating an 18% increased relative risk of CHD in individuals with elevated APOC3 levels [66,67].

In healthy individuals, APOC3-containing HDL comprises approximately 6%–13% of total HDL [68,69]. This subtype also includes electronegative HDL (H5), which is highly expressed in inflammatory states, such as in Alzheimer’s disease, and is associated with enhanced inflammatory responses in cells [70,71]. Specifically, APOC3-containing HDL has been linked to increased carotid intima-media thickness (cIMT), a key marker of cardiovascular risk. Interestingly, HDL lacking APOC3 shows opposing trends in APOA-I expression. This has led to a hypothesis that the neutral association of total APOA-I with cIMT results from the presence of two HDL subtypes with counteractive effects. Furthermore, when adjusted for TG and total APOB levels, the association between APOC3 and cIMT is no longer significant, whereas APOC3 within HDL remains significantly associated, suggesting its independent impact on cIMT [72]. Another critical feature of APOC3-containing HDL is its extended circulatory half-life, approximately 2.5 days longer than HDL lacking APOC3, potentially accelerating the progression of CVD [73]. The function of APOC3 within HDL is believed to depend on the adenosine triphosphate-binding cassette transporter A1 (ABCA1) pathway, which facilitates cholesterol efflux from macrophages. Studies with ABCA1-deficient mice (ABCA1−/−) have shown impaired formation of APOC3-containing HDL, resulting in APOC3 accumulation primarily on VLDL particles [74,75]. Reduced ABCA1 activity, as observed in T2DM, limits cholesterol transport from cells to small HDL particles, contributing to hypertriglyceridemia even at low plasma APOC3 levels, emphasizing the importance of ABCA1 in lipoprotein metabolism and cardiovascular health [76]. In individuals with *APOC3* variants, an increase in HDL-C levels coincides with lifelong reductions in plasma TG and APOC3 levels, which are linked to lower CHD risk (Figure 3).

Regarding HDL composition, HDL particles containing apolipoprotein E (APOE) or APOC3 display elevated phosphatidylcholine levels [77], a substrate for lecithin-cholesterol acyltransferase (LCAT). This elevated substrate availability may increase LCAT activity in these subtypes. Despite high phosphatidylcholine levels, LCAT activity and cholesterol transport efficiency may be limited by the presence of APOC3. Studies have shown that APOC3 reduces LCAT activity, impacting HDL’s capacity for cholesterol esterification and transport, ultimately impairing HDL functionality and affecting cholesterol metabolism and cardiovascular health [77].

Cholesteryl ester transfer protein (CETP) plays a key role in mediating cholesterol and TG exchanges between HDL, VLDL, and LDL. Studies suggest that CETP inhibition could increase HDL-C levels and reduce CVD risk [78,79,80,81,82]. However, research involving CETP inhibitors, such as torcetrapib and evacetrapib, has produced mixed results regarding their cardiovascular benefits. These inhibitors were found to increase APOC3-containing HDL, a subtype associated with higher CVD risk [83]. In the ACCELERATE trial, which evaluated the effects of evacetrapib in 12,092 patients at high CVD risk, CETP inhibition resulted in a 133.2% increase in HDL-C and a 31.1% decrease in LDL-C compared to the placebo group. However, these lipid changes did not yield a reduction in cardiovascular events [84]. Consequently, focusing on lipoproteins that are directly implicated in CVD, rather than solely increasing HDL-C through CETP inhibition, may be a more effective therapeutic approach.

#### 3.2.2. Low-Density Lipoprotein

APOC3 is increasingly recognized as a key factor associated with atherosclerosis despite its presence in LDL in small amounts [85,86,87]. In the typical lipoprotein cycle, APOC3 found on LDL primarily originates from APOC3 on VLDLs through metabolic processes [88]. Mendivil et al. highlighted that APOC3 not only participates in metabolic cycling but is integral to regulating lipid metabolism rates in physiological circulation [89]. Additionally, a study by Zheng et al. [84] revealed that APOC3-containing TRLs lacking apoE undergo faster metabolism, with many of these particles ultimately converting to LDL. In contrast, TRLs with apoE undergo rapid clearance, reducing LDL formation. This differentiation in APOC3 and apoE’s roles underscores their distinct impacts on TRL metabolism and LDL generation, affecting cardiovascular health risks [87]. Importantly, APOC3-lacking LDL does not promote adhesion molecule production in endothelial cells or damage these cells. In contrast, APOC3-containing LDL adversely affects endothelial function (Figure 4) [86]. Mendivil et al. estimated that approximately 10%–20% of LDL particles carry APOC3 [89], with APOC3-containing TRLs representing the primary precursors to LDL. In the conversion process, approximately 70% of APOC3-containing TRLs ultimately transition to LDL, forming the following two subtypes: E2CIII1 LDL (20%) and E2CIII2 LDL (50%) [87]. Regression analysis has shown that APOC3-containing LDL elevates CVD risk by an estimated 138% compared to non-APOC3-containing LDL [89]. Recent findings suggest that electronegative LDL (L5) presents an even greater CVD risk than other LDL subtypes, promoting inflammation and atherosclerosis progression [90,91,92]. L5, an LDL subtype with significant APOC3 content, has emerged as a focal point in CVD risk assessment and prevention strategies [93]. The mechanisms underlying the impact of APOC3-containing LDL on cardiovascular health remain to be fully elucidated; however, evidence suggests that it contributes to oxidative stress-induced endothelial damage and senescence, mediated through the reactive oxygen species/F-box only protein 31 (FBXO31) signaling pathway [94]. Studies have further observed increased serum levels of APOC3-containing LDL in high-fat diet-fed hamsters, suggesting a possible dietary influence [94]. In diabetes research, APOC3 has been shown to influence LDL binding. Importantly, the apoB protein in LDL contains a specific binding site (site A) that interacts with proteoglycans such as biglycan [86]. The presence of APOC3 modifies LDL structure, enhancing its propensity to bind proteoglycans, potentially leading to LDL accumulation in vessel walls. This accumulation activates endothelial and immune cells, which release inflammatory factors, exacerbating vascular damage and disease progression. Notably, of the three isoforms of APOC3, only apoCIII2 within LDL has been implicated in triggering inflammatory responses in patients with diabetes, including the production of interleukin (IL)-6, TNF-α, and IL-8 [86]. This finding underscores the pivotal role of apoCIII2 in vascular inflammation and highlights its impact on cardiovascular health and diabetes complications. Studies have also linked less-dense LDL, frequently elevated by high-carbohydrate diets, to slower VLDL clearance—a process partially influenced by APOC3 metabolism. Additionally, APOC3 glycoform sialylation alters its metabolism, such that it dissociates from less-dense LDL density patterns and elevated TG levels due to its faster clearance rate [95].

### 3.3. APOC3-Associated Cellular Signaling

#### 3.3.1. Endothelial Cells

Functional damage of arterial endothelial cells is the key process causing atherosclerosis [96]. Human experimental studies have revealed that APOC3 is involved in atherosclerotic CVDs [97]. Inflammation of vascular endothelial cells is an important developmental cause of atherosclerosis, leading to endothelial cell dysfunction, which consequently attracts monocytes to the subendothelial space of the endothelial layer [98]. Monocytes differentiate into macrophages, which phagocytose oxidized LDL. Foam cells are formed by macrophages and accumulate in the intimal layer of the vascular wall [99]. Therefore, together with the proliferation and repair of smooth muscle cells, early atherosclerotic plaques are formed. As the sclerotic plaque gradually becomes larger, the lumen of the blood vessel narrows, resulting in insufficient blood supply and ischemic symptoms, which is known as “atherosclerotic CVD” [100]. In the inflammatory signaling cascade of NF-κB, APOC3 upregulates the expression of adhesion factors, such as intercellular adhesion molecules and VCAM [101]. It also promotes the adhesion of endothelial cells and monocytes [102]. However, treatment with statins reduces the inflammatory response and endothelial cell adhesion [103]. Under normal settings, inactivated endothelium cells exhibit significantly low levels of junctional adhesion molecule 1 (JAM-1) [104]. It has been found that APOC3 can increase the inflammatory factor TNF-α and subsequently stimulate the phosphatidylinositol 3-kinase signaling pathway, causing IκB kinase degradation and p-IκB⍺ and NF-κB signaling, resulting in the overexpression of JAM-1, also known as the F11 receptor [105]. The F11 receptor is a cell adhesion molecule that selectively localizes to the tight junctions of endothelial cells [105]. According to Heiss et al., endothelial nitric oxide (NO) synthase (eNOS) is a key regulator of cellular processes that are necessary for preserving endothelium homeostasis [106]. Reduced eNOS activation and NO release into the medium are the results of the actions of APOC3 [107]. FBXO31 plays an important role in cellular senescence and has properties similar to those of a tumor suppressor [108]. APOC3-rich LDL induces oxidative stress and expression of cellular senescence factors, such as p53/p21 and FBXO31/p-murine double minute 2, resulting in the aging of human aortic endothelial cells [94] (Figure 5, Table 1).

#### 3.3.2. Monocytes and Immune Response

Although the precise alterations in monocytes and macrophages in patients with diabetes are still mostly unclear, they usually contribute to the inflammatory milieu brought on by the disease [109]. However, the following mechanisms explain how monocytes and macrophages contribute to the development of atherosclerosis: (1) increased recruitment of monocytes to lesions; (2) increased activation of the immune system; (3) altered lipid accumulation and metabolism of macrophages; and (4) increased macrophage death [110]. Together with apoptosis-associated speck-like protein, which contains a caspase recruitment domain and pro-caspase 1, cytoplasmic nucleotide-binding oligomerization domain-like receptor protein 3 (NLRP3) forms an inflammasome [111]. According to Kim et al., the cleavage of pro-caspase 1 to active caspase 1 ultimately causes an increase in IL-1β and IL-18, which triggers immune cell pyroptosis [112]. Kidney disease is pathogenesis driven by NLRP3-inflammasome-induced inflammation, and its activation exacerbates inflammation and subsequently leads to fibrosis [113]. One of the stages in the development of atherosclerosis is the large buildup of inflammatory monocytes in the blood vessel walls [98]. APOC3 activates human monocytes [114]. Moreover, it stimulates the expression of inflammatory factors (p-p65, p-p38, p-42/44 ERK, and p-SAPK/JNK) and induces dimerization of toll-like receptors 2 and 4 in an NLRP3- and caspase 8-dependent manner to impede endothelial regeneration and promote renal injury in vivo [115] (Figure 4). Recently, it was found that guanidinylated APOC3 (gAPOC3) was strongly expressed in patients with chronic kidney disease, and gAPOC3 increased IL-1, IL-6, and superoxide levels in monocytes [116]. APOC3-containing LDL increases monocyte (human acute monocytic leukemia cell line) adhesion to endothelial cells through protein kinase C-ι- and RhoA-mediated β1-integrin activation [117] (Figure 5, Table 1).

#### 3.3.3. Pancreas β-Cells

In T2DM, insulin resistance impairs glucose uptake, resulting in high blood glucose levels [57]. However, in cases of a dysregulated immune system, the released inflammatory factors cause apoptosis of islet β-cells, which leads to insufficient insulin secretion [118]. Therefore, a reduction in the number of β-cells is one of the symptoms of diabetes. APOC3 is known to cause high TG levels, which is the reason for the increased incidence of CVD during diabetes. However, the association between APOC3-induced hypertriglyceridemia and islet β-cells remains to be elucidated. In a study that used multivariate regression analysis to evaluate patients without diabetes having systemic lupus erythematosus with blood glucose levels <110 mg/dL, a significant association was found between APOC3 and C-peptide (β coefficient, 0.27; 95% CI, 0.03–0.51 ng/mL; *p* = 0.030) [119]. Experiments conducted in APOC3 transgenic mice revealed that despite the presence of hypertriglyceridemia, insulin resistance and β-cell dysfunction were not observed [120]. In a cell culture study, APOC3 was found to stimulate islet β-cell hyperactivation of voltage-gated Ca^2+^ channels (CaV), whereas it slowed CaV hyperactivation through scavenger receptor class B, type 1/β1 integrin-dependent coactivation of PKA and Src [121] (Figure 5, Table 1).

**Table 1 ijms-25-12759-t001:** Current APOC3-associated cellular signaling pathways.

Pathway Name	Key Molecular Players	Cell Types Involved	Biological Effects (e.g., Oxidative Stress, Apoptosis, Inflammation)	Relevance to Diabetes-Related CVD	References
**NF-κB Inflammatory signaling in endothelial cells**	TNF-α, IκB kinase, p-IκBα, NF-κB, JAM-1, F11 receptor, eNOS	Endothelial cells, smooth muscle cells, monocytes	Increased expression of adhesion molecules (ICAM, VCAM) and endothelial-monocyte adhesion, inflammation, oxidative stress, reduced eNOS activation, decreased NO release, endothelial dysfunction, aging of aortic endothelial cells due to oxidative stress	APOC3-mediated endothelial dysfunction and plaque formation contribute to atherosclerosis in diabetes, impairing vascular function and exacerbating CVD	[94,96,97,98,99,102,103,105,106]
**NLRP3 inflammasome activation in monocytes**	NLRP3, caspase 1, IL-1β, IL-18, p-p65, p-p38, p-42/44 ERK, p-SAPK/JNK, TLR2, TLR4	Monocytes, macrophages, endothelial cells	Inflammation through increased cytokine production (IL-1β, IL-18), activation of inflammatory pathways (p-p65, p-p38, p-42/44 ERK), impaired endothelial regeneration, renal injury	APOC3 promotes inflammation in monocytes, contributing to plaque buildup and vascular damage, accelerating atherosclerosis	[109,111,112,113,114,116,117]
**APOC3-induced β-cell dysfunction and hypertriglyceridemia**	APOC3, voltage-gated Ca^2+^ channels (CaV), PKA, Src, scavenger receptor class B type 1/β1 integrin	β-cells, pancreatic islet cells	Dysregulation of Ca^2+^ channel activation in β-cells, hypertriglyceridemia without β-cell dysfunction, impaired insulin secretion due to inflammatory factors and β-cell apoptosis	APOC3-induced triglyceride elevation promotes systemic inflammation, β-cell dysfunction, and accelerates cardiovascular disease risk	[57,118,119,120,121]

This table summarizes the key pathways, molecular players, cell types, biological effects, relevance to diabetes-related cardiovascular diseases (CVDs), and associated references.

## 4. In Vivo Models of APOC3-Related Dyslipidemia and Cardiovascular Disease

Studies using transgenic and knockout mouse models have been pivotal in elucidating the role of APOC3 in lipid metabolism and CVDs, providing translational insights relevant to human cardiovascular risk, particularly regarding TG metabolism [120,122].

APOC3 is a central regulator of TG metabolism, exerting its influence primarily by inhibiting LPL, which slows the hydrolysis of TG. In transgenic mice that overexpress human APOC3, the resulting elevation of TRLs such as VLDL induces hypertriglyceridemia, closely mirroring the dyslipidemia observed in humans and its associated cardiovascular risks due to TG accumulation in the bloodstream [120,122]. Conversely, APOC3 knockout mice, which lacked *APOC3*, exhibit reduced TG levels (hypotriglyceridemia) due to enhanced LPL activity, facilitating efficient TG clearance. These knockout models show a favorable cardiovascular profile, marked by fewer arterial plaques and reduced atherosclerosis risk—findings that parallel human studies showing that loss-of-function APOC3 mutations are associated with lower TG levels and reduced cardiovascular risk [120,122]. However, even in transgenic mice overexpressing APOC3, the development of fatty liver was not exacerbated by the overexpression of APOC3. Despite the increase in body weight and a 10-month high-fat diet intervention, the severity of fatty liver remained unchanged, indicating that APOC3 overexpression does not directly promote the progression of fatty liver [123]. While APOC3-overexpressing transgenic mice develop hypertriglyceridemia, this condition alone does not induce β-cell dysfunction or disrupt glucose homeostasis. Such models retain normal β-cell mass, glucose-stimulated insulin secretion, and insulin sensitivity, indicating that hypertriglyceridemia alone is insufficient to cause β-cell failure in the absence of additional metabolic stressors such as hyperglycemia or insulin resistance. These observations align with human studies, reinforcing the role of APOC3 in lipid regulation and cardiovascular risk without a direct causal link to β-cell dysfunction in isolation [120]. In addition to murine models, APOC3 knockout rabbit models provide further evidence of the role of APOC3 in lipid metabolism. These models exhibit a 50% reduction in plasma TG levels, accompanied by reductions in TC, VLDL, and LDL-C levels; suppression of inflammatory responses; and prevention of atherosclerosis [6,124]. This underscores the therapeutic potential of targeting APOC3 to manage lipid imbalances and improve cardiovascular health. In contrast, transgenic models of APOC3 overexpression not only display elevated plasma TG levels [114] but also show exacerbated diabetic nephropathy [21], broadening our understanding of the systemic roles of APOC3 and its implications in metabolic diseases. Although APOC3 overexpression in mice raises plasma TG levels, this alone does not explain its cardiovascular effects. APOC3 likely acts through lipoproteins, altering their composition and lipid dynamics, which may contribute to the development of CVD. In conclusion, APOC3 transgenic and knockout models underscore the essential role of APOC3 in TG metabolism and its strong association with cardiovascular risk. Although hypertriglyceridemia contributes to CVD, it does not independently precipitate β-cell dysfunction, highlighting the complex metabolic interplay and the therapeutic potential of APOC3 as a target in managing CVD risks and metabolic disorders.

## 5. Management

Multiple studies have confirmed that APOC3 plays an important role in diabetes-related CVD. Its main functions include regulating TC levels, promoting inflammatory responses, affecting lipid metabolism, and interfering with lipoprotein clearance. However, more research is needed on the impact of reducing APOC3 in relation to diabetes-related CVD.

### 5.1. Monitoring Blood Glucose

Liraglutide, a glucagon-like peptide-1 receptor agonist, is widely used to treat T2DM. It effectively improves glycemic control by enhancing insulin secretion in response to elevated blood glucose, inhibiting glucagon release during hyperglycemic and euglycemic states, slowing gastric emptying, and reducing calorie intake and body weight. A study on patients with T2DM demonstrated that liraglutide significantly lowers the secretion rate and total levels of APOC3 [57]. The glycemic response to liraglutide therapy is positively correlated with the reduction in APOC3 secretion rates, suggesting its potential impact beyond blood glucose control. Additionally, Bozzetto et al. found that while APOC3 levels are not influenced by BMI differences, they are linked to early glucose metabolism dysregulation independent of obesity or genetic predisposition, positioning APOC3 as a potential early marker for glucose metabolism abnormalities [125].

### 5.2. Alcohol Consumption

Excessive alcohol intake significantly impacts atherosclerosis management, and a notable correlation exists between alcohol consumption and increased APOC3 levels. A large-scale study revealed that higher alcohol consumption corresponds with increased total APOC3 levels, particularly the previously described APOC3-containing HDL. Specifically, each additional alcoholic drink per week increases blood APOC3 levels by approximately 0.5% [126].

### 5.3. Exercise

Exercise is associated with reduced TGs and APOC3 levels; however, further large-scale randomized controlled trials are necessary to confirm the extent of these effects. Meta-analyses of aerobic exercise in patients with T2DM indicate that such physical activity improves glycemic control, with an average fasting blood glucose reduction of −5.12 mg/dL (95% CI: −7.78 to −2.45 mg/dL), reflecting enhanced insulin sensitivity and stabilized blood glucose [127]. Among patients with CHD, aerobic exercise has been shown to decrease TG and APOC3 levels. Given the association between elevated APOC3, dyslipidemia, and heightened cardiovascular event risk, exercise-based interventions may help reduce APOC3 levels, contributing to cardiovascular health improvements, reduced atherosclerosis, and lower heart disease risk [128]. Although exercise can lower plasma TG levels, the specific mechanisms remain unclear. LPL is one of the key factors regulating plasma TG levels [129]. However, studies on healthy humans have demonstrated that exercise does not significantly affect LPL activity [129]. On the other hand, research by Seip et al. indicates that exercise has a minor impact on LPL activity in adipose tissue, possibly due to differences in the regulatory mechanisms of LPL expression in adipose tissue compared to other tissues [130]. Therefore, the role of APOC3, as a regulator of LPL, warrants further investigation in this context. In summary, while existing evidence strongly supports the role of aerobic exercise in enhancing blood glucose regulation and lipid metabolism, further studies are essential to clarify the specific effects of varying exercise intensities, types, and frequencies on these physiological markers. Such research will strengthen the evidence base guiding clinical treatment and preventive strategies for T2DM and CVD.

### 5.4. Dietary Management

Diet plays a significant role in modulating APOC3 expression and function. Diets high in saturated fats, refined carbohydrates, and sugars have been shown to elevate the APOC3 levels, contributing to hypertriglyceridemia and promoting lipid abnormalities associated with atherosclerosis [131,132,133]. In contrast, diets rich in monounsaturated fatty acids (such as olive oil) or omega-3 fatty acids (such as fish oil) have been demonstrated to reduce APOC3 expression, leading to improvements in lipid metabolism and a reduction in the cardiovascular risk [134]. High intake of saturated fats activates specific transcription factors, such as PPAR-α, which increases APOC3 expression [135,136]. These transcription factors play a crucial role in lipid biosynthesis and directly regulate the expression of genes involved in lipid metabolism, including APOC3. These dietary effects are particularly important in patients with diabetes, as hyperglycemia exacerbates lipid metabolic abnormalities and increases the risk of CVDs [137]. Therefore, dietary modifications, particularly increasing the intake of monounsaturated and omega-3 fatty acids, should be considered as an effective strategy for reducing the cardiovascular risk in individuals with diabetes. Studies have shown that unsaturated fats in the diet significantly impact the metabolism of HDL, particularly in relation to APOC3. The presence of APOC3 inhibits the process of reverse cholesterol transport (RCT), which plays a protective role in reducing atherogenesis and CHD [138]. When HDL contains APOC3, the effectiveness of RCT is diminished, which may explain why HDL with APOC3 is associated with dyslipidemia, obesity, and CHD [138].

### 5.5. Clinical Advances: Effects of APOC3-Targeting Therapies on Cardiovascular Risk in Diabetes

#### 5.5.1. Volanesorsen

Volanesorsen, developed by Ionis Pharmaceuticals (Carlsbad, CA, USA) and granted orphan drug status by the European Medicines Agency in February 2014, is currently in phase III of clinical trials [139]. A meta-analysis [140] demonstrated that volanesorsen significantly reduced TGs (mean percentage difference, −78.85%; 95% CI, −96.04 to −61.65; *p* = 0.67; heterogeneity [I^2^] = 0%) and APOC3 (mean percentage difference, −80.08%; 95% CI, −90.02 to −71.54; *p* = 0.25; I^2^ = 29%) levels while increasing HDL-C and LDL-C concentrations. However, the therapy did not sufficiently target all atherogenic lipoproteins [140]. Thrombocytopenia has emerged as a potential adverse effect in clinical trials [141]. In a study on patients with T2DM, volanesorsen was effective in reducing plasma APOC3 and TG levels while increasing HDL-C [58]. Short-term treatment showed improvements in glucose metabolism, insulin sensitivity, and diabetes-related biomarkers, although LDL-C levels remained unaffected. Therefore, further studies are needed to confirm whether these improvements contribute to a reduction in cardiovascular event risk [58].

#### 5.5.2. Olezarsen

Olezarsen, formerly known as AKCEA-APOCIII-LRx, is an N-acetylgalactosamine-conjugated antisense oligonucleotide specifically designed to inhibit APOC3 synthesis in the liver. The drug received investigational approval from the Food and Drug Administration on 31 January 2023 [142]. In clinical studies of hypertriglyceridemia, treatment with olezarsen at doses of 200–500 mg/dL resulted in a TG reduction of approximately 60% [142]. Other studies have reported even more substantial reductions, with TG levels dropping as much as 73% within 3 months (from a baseline of 189 mg/dL to 53 mg/dL) [143].

#### 5.5.3. Plozasiran

Plozasiran, a GalNAc-conjugated siRNA targeting APOC3, effectively inhibits its production, thereby enhancing TG metabolism and reducing its levels. Clinical trials show a strong dose-dependent effect, with a placebo-adjusted reduction of −57% (95% CI, −71.9% to −42.1%; *p* < 0.001) in TG levels and a decrease of −77% (95% CI, −89.1% to −65.8%; *p* < 0.001) in APOC3 levels [144].

Meta-analyses indicate that these three APOC3-targeting agents have shown significant efficacy across various clinical trial phases (Phases I–III), achieving an average TG reduction of 57% compared to placebo, with reductions ranging from 23.7% to 94.1% [145]. These findings highlight the potential of these novel therapies in managing severe hypertriglyceridemia, particularly for patients who respond inadequately to conventional treatments. However, further studies are needed to understand how these treatments impact T2DM progression and CVD risk, emphasizing the need for more research on the metabolic and pathological implications of APOC3 inhibition.

#### 5.5.4. Additional Therapeutics

APOC3 has emerged as a promising therapeutic target, given its crucial role in TG metabolism. While some lipid-lowering agents, such as statins [146], omega-3 carboxylic acids [147], fibrates [148], and niacin [34], can reduce APOC3 levels, the effect is generally limited, and the underlying mechanisms remain complex. Therefore, more targeted and efficient APOC3-inhibiting therapies are being actively pursued for comprehensive TG management and cardiovascular risk reduction in patients with diabetes. Table 2 presents the summaries of current therapeutic strategies that effectively lowered APOC3 levels in patients with hypertriglyceridemia, providing promising treatment options for those with diabetes and elevated triglyceride (TG) levels.

## 6. Conclusions

APOC3 plays a complex role in diabetes-associated CVDs, affecting lipid metabolism, vascular health, and islet cell function through its presence in diverse lipoproteins, including VLDL, intermediate-density lipoprotein, LDL, and HDL. While APOC3-lowering drugs hold potential, their effectiveness in reducing CVD risk remains uncertain. This review underscores the need for more detailed research into the structural and functional variations in APOC3 across lipoproteins, which could uncover specific pathways linking APOC3 to CVD. A deeper understanding of these mechanisms is essential for developing precise, targeted therapies to mitigate cardiovascular risk in patients with diabetes, ultimately enhancing clinical management strategies.

## Figures and Tables

**Figure 1 ijms-25-12759-f001:**
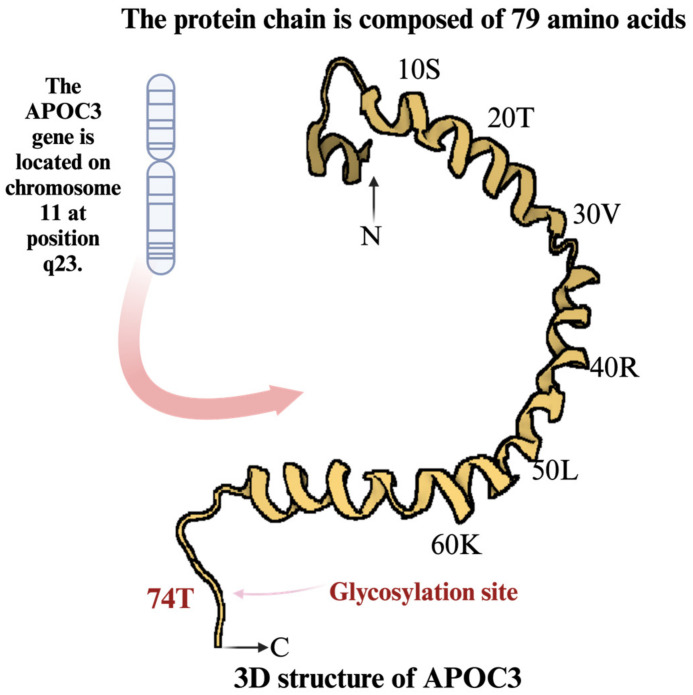
Three-dimensional structure and molecular features of apolipoprotein C3 (APOC3). The three-dimensional structure of APOC3, modeled from the protein data bank (PDB ID: 2JQ3), is depicted in its complete conformation. The genomic locus of the *APOC3* gene is shown on human chromosome 11 at position 11q23. The amino acid sequence of APOC3 is provided, with glycosylation sites highlighted to indicate specific positions where post-translational modifications occur.

**Figure 2 ijms-25-12759-f002:**
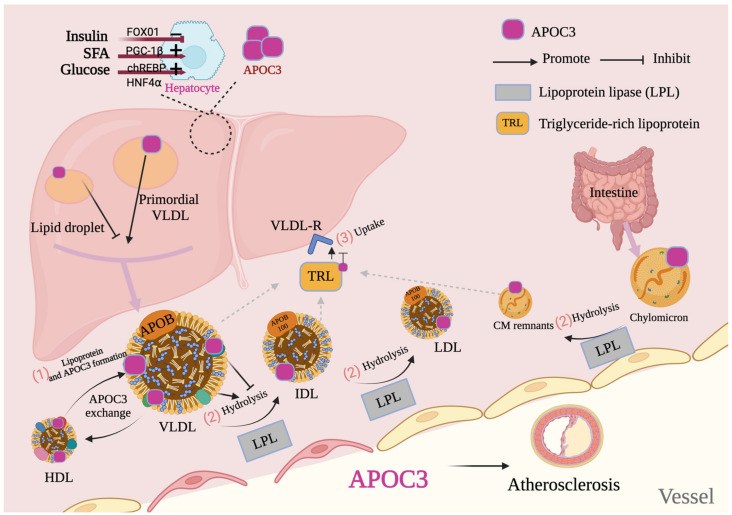
Interaction between liver-synthesized apolipoprotein C3 (APOC3) and lipoproteins. (1) These processes are related to the de novo synthesis of very low-density lipoprotein (VLDL) in the liver, absorption and secretion of chylomicrons (CM) in the intestine, secretion of peripheral tissues (lipoprotein lipase [LPL] [44]), and clearance rate of the liver. Fluctuations in yield or clearance lead to the accumulation of triglyceride (TG)-rich lipoproteins (TRLs), which results in high TRLs. (2) During LPL-promoted hydrolysis of VLDL and TGs contained in CM, APOC3 is transferred from VLDL to high-density lipoprotein (HDL) particles and subsequently back to other newly secreted VLDL. (3) TRLs are recycled from the VLDL receptor (VLDL-R) in the liver. Saturated fatty acids (SFAs) and carbohydrates (glucose and fructose) stimulate the production of APOC3 in the liver; however, insulin inhibits APOC3 expression. Abbreviations: IDL, intermediate-density lipoprotein; HNF4α, hepatocyte nuclear factor 4α; PGC-1α, peroxisome proliferator-activated receptor-γ coactivator 1α; ChREBP, carbohydrate response element-binding protein; APOB, apolipoprotein B. Red arrow, stimulation; red line with end bar, inhibition; purple arrow, direction or flow.

**Figure 3 ijms-25-12759-f003:**
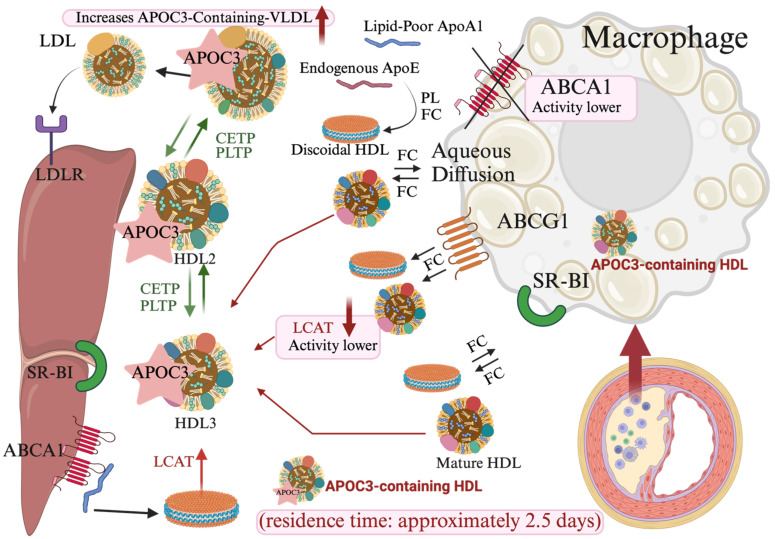
Apolipoprotein C3 (APOC3)-containing high-density lipoprotein (HDL): impacts on cholesterol transport and cardiovascular health. The reverse cholesterol transport pathway facilitated by HDL is essential for maintaining cholesterol homeostasis and promoting cardiovascular health. In this process, HDL particles gather excess cholesterol from peripheral tissues and deliver it to the liver for excretion. However, APOC3-associated HDL (APOC3-HDL) has been shown to disrupt this beneficial mechanism. APOC3 inhibits lipoprotein lipase (LPL) activity and may impede cholesterol esterification, thereby impairing HDL maturation and functionality. APOC3-HDL has an extended circulation time, averaging approximately 2.5 days, which could contribute to an accelerated risk of cardiovascular disease. Furthermore, adenosine triphosphate-binding cassette transporter A1 (ABCA1) is essential for the formation of APOC3-HDL, and reduced ABCA1 activity can lead to intracellular cholesterol accumulation, exacerbating hypertriglyceridemia. Although HDL typically provides cardiovascular protection, APOC3-HDL is associated with heightened cardiovascular disease risk, underscoring the detrimental effects of APOC3 on cholesterol transport and cardiovascular health. Abbreviations: LCAT, lecithin-cholesterol acyltransferase; CETP, cholesteryl ester transfer protein; LDL, low-density lipoprotein; LDL-R, low-density lipoprotein-receptor; ApoE, apolipoprotein E; ApoA1, apolipoprotein A1; SR-BI, scavenger receptor class B, type 1; ABCG1, ATP-binding cassette subfamily G member 1; PL, phospholipid; FC, phosphatidylcholine; PLTP, phospholipid transfer protein.

**Figure 4 ijms-25-12759-f004:**
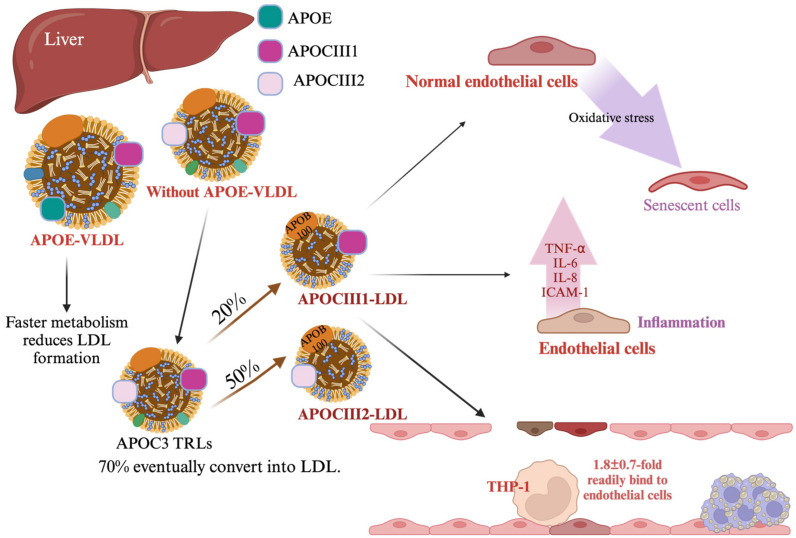
Cardiovascular risk associated with APOC3-containing low-density lipoprotein (LDL). LDL particles containing APOC3 substantially elevate the risk of cardiovascular disease compared to those lacking APOC3, with studies estimating an increased risk of approximately 138%. APOC3-containing LDL originates primarily from triglyceride-rich lipoproteins (TRLs) that lack apoE, which metabolize faster than TRLs containing apolipoprotein E (apoE). TRLs without apoE may lose part or all of their APOC3 during their conversion into LDL, while apoE presence facilitates their rapid clearance. Additionally, APOC3 influences the binding properties of LDL, primarily through interactions between the apolipoprotein B (apoB) protein on LDL and specific proteoglycan binding sites, such as site A, which interacts with biglycan. This binding propensity may promote LDL retention within the vascular wall, enhancing inflammation and cardiovascular risk. Abbreviations: APOC3, apolipoprotein C3; APOB, apolipoprotein B; VLDL, very low-density lipoprotein; IL-6, interleukin-6; TNF-α, tumor necrosis factor α; IL-8, interleukin-8; ICAM-1, intercellular adhesion molecule-1. Black arrow, red arrow, and purple arrow, direction; pink arrow, flow.

**Figure 5 ijms-25-12759-f005:**
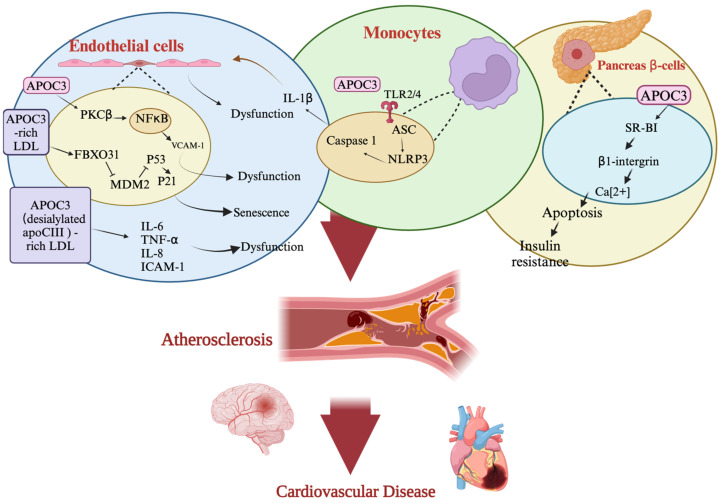
Impact of apolipoprotein C3 (APOC3) on endothelial cells, monocytes, and pancreas β-cells in diabetes and cardiovascular diseases. APOC3 acts as a pro-inflammatory factor closely associated with diabetes-related cardiovascular disease. APOC3 initiates atherosclerotic processes in endothelial cells and monocytes but disrupts glucose regulation in pancreatic β-cells. Abbreviations: LDL, low-density lipoprotein; PKCβ, protein kinase Cβ; FBXO31, F-box only protein 31; MDM2, mouse double minute 2 homolog; TLR2/4, toll-like receptor 2/4; NLRP, nucleotide-binding oligomerization domain, leucine-rich repeat and pyrin domain-containing family; VCAM, vascular cell adhesion molecule; IL, interleukin; ASC, apoptosis-associated speck-like protein; SR-BI, scavenger receptor class B, type 1; TNF-⍺, tumor necrosis factor-⍺; NF-κB, nuclear factor κappa B; ICAM1, intercellular adhesion molecule-1. Black arrow, stimulation; red arrow, direction or flow; black line with end bar, inhibition.

**Table 2 ijms-25-12759-t002:** Current therapeutic strategies that effectively lowered APOC3 levels in patients with hypertriglyceridemia.

Drug Name	Pathway	Indication/Targeted Population	Effect	Others	References
**Omega-3 PUFAs**	PPAR-α	Hypertriglyceridemia	APOC3 concentrations following (WMD, −22.18 mg/L; 95% CI, −31.61 to −12.75; *p* < 0.001; I^2^ = 88.24%)	EPA ethyl esters are particularly effective	[147]
**Fenofibrate**	PPAR-α	Hypertriglyceridemia Type 2 diabetes Metabolic syndrome Dyslipidemia	APOC3 concentrations following (WMD, −4.78 mg/dL; 95% CI, −6.95 to −2.61; *p* < 0.001; I^2^ = 66.87%)		[148]
**Statin**	Unknown mechanism	Hypercholesterolemia and CAD Obese men Type 2 diabetes mellitus	APOC3 concentrations following (WMD, −2.71; 95% CI, −3.74 to −1.68; *p* < 0.001; I^2^ = 73.83%)		[146]
**Olezarsen (phase III)**	Antisense oligonucleotide targeting APOC3 mRNA	Familial chylomicronemia syndrome	Lowers APOC3 concentration up to 74%		[149]
**Volanesorsen** **(phase III)**	Antisense oligonucleotide targeting APOC3 mRNA	Familial chylomicronemia syndrome	Lower APOC3 concentration up to 80.08%	With thrombocytopenia risk	[150]
**Plozasiran** **(phase 3)**	GalNAc conjugated siRNA	Persistent chylomicronemia	Lower APOC3 concentration up to −80%		[151]

PUFAs, polyunsaturated fatty acids; APOC3, apolipoprotein C3; PPAR-α, peroxisome proliferator-activated receptor α; WMD, weighted mean difference; I^2^, percentage of total variability in a set of effect sizes due to true heterogeneity; EPA, eicosapentaenoic acid; CI, confidence interval.

## Data Availability

Not applicable.

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
