# Peer review of "Multifaceted Role of Apolipoprotein C3 in Cardiovascular Disease Risk and Metabolic Disorder in Diabetes"

_ijms, 2024, doi:10.3390/ijms252312759_

Round 1
Reviewer 1 Report (New Reviewer)
Comments and Suggestions for Authors
1. This systematic review comprehensively summarises the multifaceted role of apolipoprotein C3 (APOC3) in the risk of diabetic cardiovascular disease and metabolic disorders. However, there are still some areas for improvement. The chapter focuses on the role of APOC3 in diabetes-related cardiovascular disease (CVD), and provides a comprehensive description of its functions in different lipoproteins, related gene variants, cell signalling pathways, in vivo models, which is novel. In particular, the detailed discussion of the role of APOC3 in high-density lipoprotein (HDL) and low-density lipoprotein (LDL) provides new insights into the relationship between APOC3 and CVD.
2. It is suggested that when discussing the interaction between APOC3 and other factors (such as diet, exercise, etc.), further in-depth exploration of its potential molecular mechanisms and how these interactions affect the development of diabetes-related CVD can be conducted. In addition, for some controversial data or opinions, more in-depth analysis and discussion can be conducted to guide the reader to a more comprehensive understanding of research in this field.
3. The overall writing style is clear and the structure is clear. The abstract accurately summarizes the main content and research objectives of the article, and the keywords are appropriate.
4. Please add a Table . Current APOC3-Associated Cellular Signaling to make the article more understandable.
Author Response
Please see the attachment.

Reviewer 2 Report (New Reviewer)
Comments and Suggestions for Authors
The authors report on the multifaceted role of apolipoprotein C3 in cardiovascular disease, which mitigates the risk of diabetes and metabolic disorders, but needs to be more specific.
Minor issues
1. In p value, p is italicized (P = 0.05)
2. Why is APOC3 Italian on line 192?
3. Unify the font of title and subtitle within a sentence.
4. It would be nice to include a protein structure diagram for the APOC3 gene in the introduction.
Major issues
1. An overview of how APOC3 affects cardiometabolic should be given in the introduction and then narrowed down to how it regulates diabetes and TG to avoid confusion in understanding the manuscript.
2. Requires a complete reorganization and rewrite in the same way as #1 above
Round 2
Reviewer 2 Report (New Reviewer)
Comments and Suggestions for Authors
Good job
This manuscript is a resubmission of an earlier submission. The following is a list of the peer review reports and author responses from that submission.
Round 1
Reviewer 1 Report
Comments and Suggestions for Authors
This paper discusses the relationship between ApoC3 and diabetes and other diseases, and the relevance of ApoC3, from basic research to clinical study, including the latest ApoC3 treatment drugs.
However, this review paper is too broad in scope, making it difficult to understand the overall picture, and it is extremely unclear what the purpose and main theme of the paper is. As it stands, it appears to be a list of previously published papers. In addition, although the paper indexes the relationship between ApoC3 and each disease, it is not clear whether ApoC3 is the cause of the disease, even though it is shown to be related.
The conclusion is very vague in the last line of the introduction and the first line of the conclusions, and it is not clear what this review article is trying to say.
The authors should clarify the topic of this paper. Considering that ApoC3 is found in TG-rich lipoproteins in relation to atherosclerosis, the authors should focus on discussing TG.
By describing the results of in vivo animal studies, such as transgenic mice and knockout mice, it would be easier for readers to understand whether ApoC3 is a reasonable cause or not, and to clearly examine whether it is consistent or inconsistent with the clinical outcomes of diabetes, CVD, and atherosclerosis in humans.
Major comments
<abstract>
Line 17: Regarding the way ApoC3-induced disease is written: Is this the correct way to write it, since it has not been proven that ApoC3 causes disease?
<Introduction>.
The cited papers are generally old, and newer, more appropriate papers are needed.
・ The discussion of LDL-C and TG is mixed together, but considering that TG-rich lipoproteins are where ApoC3 is found, the discussion and description should focus on TG.
The mechanism by which TG levels rise in diabetes should be added, and it seems necessary to state as a basic premise of this story that ApoC3 levels on TG-rich lipoproteins rise as TG levels rise.
References 6 and 7 are an original article (7) and a commentary on that article (6), so it is not desirable to index them twice.
Minor
Line 134: ([41] is a misprint?; Line 392, APCO3 is a misprint?
Line 306-307: Is this an inappropriate sentence?
Line 334-338: Is this correct? Is this not the subjective opinion of the authors?
Reviewer 2 Report
Comments and Suggestions for Authors
The authors overviewed the role of APOC3 in diabetes-associated CVDs and on current preclinical studies examining the prevention of APOC3-induced diseases, such as stroke, cardiometabolic disorders, diabetes mellitus, and coronavirus disease. APOC3 and APOC3-rich lipoproteins play a major role in the development of CVDs. However, the underlying mechanism has not been clarified and needs to be explored further. Since APOC3 is a promising biomarker, the authors suggest that there is an urgent need for a high-throughput, clinically feasible method to investigate the physiological mechanisms of action of APOC3-associated lipoproteins that cause CVD in animals and humans. In this review, the authors summarized the relationship between APOC3-related genes and lipoproteins and explained the role of APOC3 in different diseases. They concluded that prospective, large population studies should be conducted, and therapeutic strategies should be developed to lower APOC3 levels in the blood.
In general, the topic is highly interesting, but the review is not well focused. Based on the text it is not obvious for the potential readers that APOC3 is a structure protein of triglyceride-rich lipoproteins, which have a key role in the atherosclerotic process. Therefore, from clinical point of view, is us useless to discuss the APOC3 elevation separately. There are many excellent reviews focusing on diabetes-associated dyslipidemias and the importance of triglyceride-rich lipoproteins. This manuscript is rather confusing.
Further comments:
· The review focuses on diabetes-associated CVD risk and APOC3. The COVID-19 infection is not a player in this process, I would delete the sections discussing the COVID-19.
· Line 168: these are not gene mutations, these are mainly single nucleotide polymorphisms, or gene variants.
· Line 257: Stroke is a disease, please write Stroke or Stroke/TIA instead of stroke events.
· Line 336. Any viral or bacterial infection can provoke ketoacidosis in type 1 diabetes, it is not a COVID-specific problem.
· Fig 4. Sesamol is not discussed in the text, there are not evidence-based data on its risk reducing efficacy, therefore, it should be deleted.
· English needs extensive editing.
Comments on the Quality of English LanguageEnglish needs extensive editing.
Round 2
Reviewer 1 Report
Comments and Suggestions for Authors
It is difficult to say that the corrections made to the points raised in the previous review are sufficient. A fundamental change in the argument is desirable.
Reviewer 2 Report
Comments and Suggestions for Authors
The manuscript was significantly improved. The authors removed the chapters focusing on COVID-19, and rephrased many other parts of the text. The reference list was updated.
Comments on the Quality of English LanguageMinor editing is needed.